# CBGRU: A Detection Method of Smart Contract Vulnerability Based on a Hybrid Model

**DOI:** 10.3390/s22093577

**Published:** 2022-05-07

**Authors:** Lejun Zhang, Weijie Chen, Weizheng Wang, Zilong Jin, Chunhui Zhao, Zhennao Cai, Huiling Chen

**Affiliations:** 1College of Information Engineering, Yangzhou University, Yangzhou 225127, China; mz120200850@yzu.edu.cn; 2Research and Development Center for E-Learning, Ministry of Education, Beijing 100039, China; 3Cyberspace Institute Advanced Technology, Guangzhou University, Guangzhou 510006, China; 4Computer Science Department, City University of Hong Kong, Hong Kong; m5232117@u-aizu.ac.jp; 5School of Computer and Software, Nanjing University of Information Science and Technology, Nanjing 210044, China; zljin@nuist.edu.cn; 6College of Computer Science and Technology, Harbin Engineering University, Harbin 150001, China; zhaochunhui@hrbeu.edu.cn; 7Department of Computer Science and Artificial Intelligence, Wenzhou University, Wenzhou 325035, China; cznao@wzu.edu.cn

**Keywords:** smart contract, security, vulnerability detection, hybrid model

## Abstract

In the context of the rapid development of blockchain technology, smart contracts have also been widely used in the Internet of Things, finance, healthcare, and other fields. There has been an explosion in the number of smart contracts, and at the same time, the security of smart contracts has received widespread attention because of the financial losses caused by smart contract vulnerabilities. Existing analysis tools can detect many smart contract security vulnerabilities, but because they rely too heavily on hard rules defined by experts when detecting smart contract vulnerabilities, the time to perform the detection increases significantly as the complexity of the smart contract increases. In the present study, we propose a novel hybrid deep learning model named CBGRU that strategically combines different word embedding (Word2Vec, FastText) with different deep learning methods (LSTM, GRU, BiLSTM, CNN, BiGRU). The model extracts features through different deep learning models and combine these features for smart contract vulnerability detection. On the currently publicly available dataset SmartBugs Dataset-Wild, we demonstrate that the CBGRU hybrid model has great smart contract vulnerability detection performance through a series of experiments. By comparing the performance of the proposed model with that of past studies, the CBGRU model has better smart contract vulnerability detection performance.

## 1. Introduction

The concept of smart contracts was introduced by Nick Szabo [1]. He describes a smart contract as “a set of agreements that are defined digitally and that contain information about how the participants will fulfill those agreements”. The purpose of proposing smart contracts was to be able to implement the content of the contracts through cryptographic protocols and digital security mechanisms. However, due to the limitation of technology at that time, there was no carrier to carry smart contracts, so smart contracts were not applied. It was not until 2008 when an academic named Satoshi Nakamoto proposed the concept of Bitcoin. The underlying technology of Bitcoin is blockchain, which can be used as the carrier of smart contracts [2]. Blockchain is essentially a decentralized distributed ledger database. Due to its tamper-evident characteristic, blockchain is also applied to the Internet of Things (IoT) to ensure the security of connected data. Scholars have done a lot of research on how to improve the performance of blockchain to meet the needs of IoT; for example, Hang L et al. proposes a transaction traffic control approach based on fuzzy logic to enhance the blockchain network’s transaction-processing capacity [3]. In 2013, Vitalik Buterin proposed Ethereum [4] to formally introduce smart contracts into the blockchain platform, which has a Turing-complete and powerful scripting system that enables more advanced distributed applications based on smart contracts. A smart contract [5] is a computer protocol that is self-executing and self-verifying once deployed. Due to their characteristics, smart contracts are used in the Internet of Things. Zhang Y et al. proposed a distributed access control framework for smart cities by combining blockchain smart contract technology and the attribute-based access control (ABAC) model [6]. B. Duan et al. propose an Internet of Things (IoT) charging scheduling method based on smart contract technology [7]. Alzubi O A et al. presents a novel blockchain and artificial intelligence-enabled secure medical data transmission (BAISMDT) for IoT networks [8]. Similar to traditional contracts, the life cycle of a smart contract consists of three parts: contract generation, contract release, and contract execution. The life cycle of a smart contract is shown in Figure 1. Smart contracts will eventually be deployed to the blockchain for self-verification and self-execution. When the execution conditions set by the contract content are triggered, the smart contract can automatically output the response without relying on the cooperation of third parties.

Smart contracts have complex time and order dependencies, but errors in the logic of the contract code and errors in the text of the contract code will lead to vulnerabilities in the smart contract, which will eventually result in incorrect automatic execution. Ethereum is one of the most popular blockchain platforms [4], where tens of thousands of smart contracts are deployed, and the smart contract itself is an Ethereum account, called the external account controlling billions of dollars’ worth of Ether (cryptocurrency Ethereum). Such a huge amount of digital assets held by Ethereum is a huge attraction for attackers. In 2016, USD 55 million worth of Ether was lost due to a vulnerability in The Dao’s smart contract [9]. In 2017, over USD 30 million worth of Ether was lost due to a smart contract vulnerability in the Parity wallet [10]. Such security issues create serious obstacles to the development of blockchain and cause a crisis of trust in smart contracts among users, so an efficient smart contract vulnerability detection tool is very important and urgent. Therefore, we propose an efficient deep learning hybrid model named CBGRU for detecting smart contract vulnerabilities in this paper. Our key contributions are as follows:By combining different word embedding methods and deep learning methods, the accuracy of smart contract vulnerability detection is improved.CBGRU applies hybrid networks to smart contract vulnerability detection for the first time, and the hybrid network model proposed in this paper can detect several different smart contract vulnerabilities while maintaining good detection performance.Through extensive experiments, it is demonstrated that the CBGRU model proposed in this study combines the advantages of deep learning and hybrid learning, and has better smart contract vulnerability detection performance compared to a single neural network model.

The paper is organized as follows: We present current research on smart contract vulnerability detection and hybrid models in Section 2. We discuss our methodology in Section 3. We present the experimental procedure, which includes dataset processing and performing performance comparisons, in Section 4. Finally, we conclude the whole paper and future work in Section 5.

## 2. Related Work

### 2.1. Smart Contract Vulnerability Detection

Researchers have also come up with some smart contract vulnerability detection tools. Oyente, proposed by Luu L et al., is a smart contract vulnerability detection tool based on symbolic execution and is currently capable of detecting seven types of smart contract vulnerabilities [11]. SmartCheck is an extensible static analysis tool proposed by Tikhomirov S et al. SmartCheck converts Solidity source code into an XML-based intermediate display and checks it against XPath patterns [12]. Mythril is the official smart contract vulnerability detection tool for Ethereum, which can detect a large number of smart contract security issues, with the main idea of using symbolic execution to explore all possible insecure paths [13]. Security is a smart contract static analysis tool for detecting the security properties of smart contract EVM bytecodes [14]. Slither is also a static analysis tool for smart contract vulnerability detection [15]. ContractFuzzer is a fuzzy testing tool for smart contract vulnerabilities, capable of generating fuzzy test inputs based on the smart contract’s API specification, recording the runtime state of the smart contract through an Ethernet Virtual Machine (EVM), analyzing logs, and reporting security vulnerabilities [16]. Echidna [17] and Ethracer [18] are also fuzzy testing tools for smart contract vulnerabilities. These tools are mainly based on formal verification, symbolic execution, static analysis, taint analysis, and fuzzy testing, and rely on hard logical rules defined in advance by experts when performing vulnerability detection. However, with the development of smart contracts, these tools can no longer meet the current needs. Traditional smart contract vulnerability detection tools are inadequate for the following reasons:Smart contracts are becoming more and more complex in their structure to achieve complex functionality, and the variety of smart contract vulnerabilities is increasing, and the rules defined by experts based on vulnerabilities cannot keep up with the speed of smart contract vulnerability updates.A crude overlay of several expert-defined logic rules can lead to a high false-alarm rate, and expert rule-based smart contract vulnerability detection tools are not suitable for general smart contract vulnerability detection situations.The attacker can use techniques to bypass inspection patterns against these rules defined in advance. Expert rule-based smart contract vulnerability detection tools cannot be updated on time.

In recent years, deep learning techniques have been used in various fields, including the field of vulnerability detection. Russell R L et al. used convolutional neural networks (CNNs) for vulnerability classification, learning features through neural networks, and extracting control flow graphs (CFGs) of functions at the function level, obtaining excellent overall results [19]. Sicong Gao et al. proposed the BGNN4VD model, which performs vulnerability detection by constructing bipartite graph neural networks, and proved through experiments that BGNN4VD has a high precision and accuracy [20]. Zhen Li et al. proposed a deep learning-based vulnerability detection system, named VulDeePecker [21]; it applies bi-directional long and short memory neural networks to vulnerability detection. Researchers have also combined deep learning with smart contract vulnerability detection. Yu X et al. developed a systematic and modular framework for smart contract vulnerability, called DeeSCVHunter, proposing the concept of vulnerability candidate slicing (VCS), which contains rich semantic and syntactic features that can significantly improve the performance of deep learning smart contract models in smart contract vulnerability detection [22]. DeeSCVHunter can only detect re-entry vulnerabilities and timestamp vulnerabilities, while the CBGRU model proposed in this paper can detect multiple vulnerabilities, and DeeSCVHunter does not consider the correlation between word embedding models and deep learning models when performing word embedding and feature extraction. Wu H et al. used a smart contract representation method based on key data flow graph information for capturing the key features of contracts before performing smart contract vulnerability detection, while overfitting can be avoided during training, and proposed a new tool, named Peculiar [23]. Peculiar improves detection performance by leveraging the technique of critical data flow graphs, but the process of building critical data flow graphs is complex and Peculiar can only detect smart contract reentry vulnerability. Qian P et al. proposed to combine a bidirectional long-short memory network with an attention mechanism for smart contract vulnerability detection, named BLTM-ATT, and they demonstrated through extensive experiments that this method can achieve better accuracy [11]. The BLSTM-ATT model also does not consider the association between the word embedding model and the deep learning model. The BLSTM-ATT model can only detect smart contract reentry vulnerability. Xing C et al. proposed a new slicing matrix for detecting vulnerabilities and experimentally demonstrated that the slicing matrix can improve the accuracy of vulnerability detection [24]. Xing C et al. select “Return” as the segmentation point when segmenting the contracts opcodes, but this cannot completely distinguish between useful and useless operands, which leads to partial feature loss and thus affects the performance of the model in detecting smart contract vulnerabilities. The CBGRU model pre-processing takes into account the integrity of the smart contract to ensure the integrity of the smart contract semantics when performing feature extraction. Zhipeng Gao et al. proposed a method for automatically learning smart contract features based on character embedding and vector space comparison. The main approach is to parse smart contract code into a stream of characters with code structure information, convert the code elements into vectors, and thus compare the similarity between the coded vectors and known bugs to identify potential smart contract vulnerabilities [25]. Goswami S et al. proposed a smart contract vulnerability detection tool named TokenCheck, which was tested on the Ethernet smart contract dataset ERC-20 and obtained good results [26]. Goswami S et al. used a single LSTM neural network for feature extraction. The CBGRU model proposed in this paper combines the advantages of multiple deep learning models with more adequate feature extraction. From the above study, it can be seen that the combination of deep learning and vulnerability detection is feasible to obtain excellent vulnerability detection results, and the combination of smart contract vulnerability detection and deep learning can achieve the same good results. However, the association between word embedding models and deep learning models was not considered in previous studies, and the advantages of combining multiple deep models were not combined when extracting feature extraction.

### 2.2. Deep Learning Hybrid Models

The CBGRU model proposed in this paper is a hybrid learning model, and the role of the hybrid model is to combine the strengths of each network to achieve better results. Hybrid models have been used in many fields and have obtained great results. Du S et al. combined one-dimensional convolutional neural networks (1D-CNNs) with bi-directional long-short memory networks (Bi-LSTMs) for air quality prediction, where convolutional neural networks are used to extract features and bi-directional long-short memory neural networks are used to learn the correlation of features [27]. Fu L et al. argued that deep learning models based on convolutional networks with DNA sequences as input found limited information and the prediction would be unsatisfactory, so they proposed a new deep learning model based on hybrid sequences [28]. Xi J et al. applied hybrid networks to high-resolution image classification, combining fully connected networks, convolutional neural networks, and fully convolutional networks, and showed experimentally that hybrid integrated learning methods have better performance compared to single classical neural networks and deep learning methods [29]. Yue W et al. combined the word vector model (Word2vec), the bi-directional long-term short-term memory network (BiLSTM), and the convolutional neural network (CNN), and experimentally demonstrated that the hybrid network model outperformed the single-structured neural network in short text classification [30]. Chuang P J et al. combined the Naive Bayes model and C4.5 algorithms to improve the performance and training time of training classification models in network intrusion detection, and experimental results show that their proposed hybrid model can reduce the required training time while providing good detection performance [31]. Duan J et al. proposed a hybrid neural network model (MLCN and BiGRU-ATT) that combines the multilayer convolutional neural network (MLCNN) and the bidirectional gated recurrent unit (BiGRU) with the attention mechanism to be applied to news classification [32]. The following conclusions can be drawn from the above research.

Hybrid models have been well used in image recognition, sentiment classification, network intrusion detection, and news classification.Hybrid models have better performance compared to a single deep learning model, and the training time is faster than a single model due to the lower complexity of each model in the hybrid model.Hybrid networks are more conducive to classification because they combine the advantages of different deep learning models.

Therefore, in this paper, we choose to apply the hybrid network to the task of smart contract vulnerability detection.

### 2.3. Research Motivation

In previous deep learning-based smart contract vulnerability detection tasks, the neural network models used are linear and had a single structure, and the training process for the models is similar. The training process has four steps.

Determining the label of the training data.Pre-processing of training data and making changes to the form of training dataExtracting the feature values by deep learning modelsClassification of training data by the classifier of the model

The process is shown in Figure 2.

The single deep learning model can be achieved by increasing the number of layers of the model to obtain a higher accuracy of classification, but it brings the problem that the complexity of the model will also grow, and problems such as overfitting and a long training time will occur. The emergence of hybrid networks can improve such problems. Hybrid models perform feature extraction by two deep learning models, and then the extracted feature values are weighted so that the extracted feature values are more adequate. The hybrid model training process is shown in Figure 3.

Before a deep learning model can be used to extract the features, the data needs to be processed so that it fits the input of the neural network. In previous research, word embedding has been used to convert smart contracts into multidimensional matrices. However, the connection between deep learning methods and word embedding methods has not been considered in previous studies. Each deep learning method and word embedding method has its advantages and disadvantages, so combining the advantages of deep learning models with the advantages of word embedding models is the key to improving the accuracy of smart contract vulnerability detection. Each deep learning method has different characteristics for solving different types of tasks; on the other hand, each word embedding method has its advantages and disadvantages, it is of great importance to combine the optimal word embedding method and the optimal deep learning model for smart contract vulnerability detection tasks. Since each word embedding approach may be incomplete in its representation of smart contract features, the proposed solution to this problem is to combine the advantages of different word embedding approaches and deep learning models, so we propose a new hybrid deep learning model named CBGRU after discussing various neural networks and word embedding approaches. The model uses two different word embedding methods for word embedding and two different deep learning models for the feature extraction phase. In addition, a large number of classification experiments are conducted in this study, which is used to demonstrate the excellent performance of CBGRU in smart contract vulnerability detection.

## 3. Overall Framework

The core idea of this paper is to combine the advantages of different deep learning models with models of different word embeddings to improve the performance of smart contract vulnerability detection. The framework of the hybrid deep learning network proposed in this paper is shown in Figure 4.

As can be seen in Figure 4, the proposed framework of this paper is divided into four parts.

Pre-processing of the dataset.Mapping high-dimensional smart contracts to low-dimensional vectors via word embedding models.Extract the feature values by two neural networks, then concatenate the feature values.Performing classification and deriving results.

The research in this paper focuses on the second and third parts, combining the advantages of different word embedding methods and different depth models to improve the performance of smart contract vulnerabilities. In Part two and Part three, we discuss the currently popular word embedding methods, Word2Vec and FastText, and the currently commonly used deep learning models GRU, CNN, BiLSTM, LSTM, and BiGRU.

### 3.1. Overall Model Structure

Each deep learning model has its advantages and disadvantages in extracting features, for example, CNN (Convolutional Neural Networks) has strong feature extraction and generalization ability, but the performance is average for context-dependent information. RNN (Recurrent Neural Network) has a good feature extraction ability for information with sequential dependence. However, RNNs are less capable of extracting features for data with long-term dependencies. LSTM (Long Short-Term Memory) can solve the long time-dependency problem of RNN [33], but it takes too long to perform feature extraction and the effect of the feature extraction is not as good as CNN. The characteristics of each word embedding method are also different. As can be seen in Figure 4, the structure of the deep learning model used in this paper is composed of two branches, and the deep learning model of each branch condenses the feature values after extracting them so that the extracted feature values are more adequate compared to the neural network structure of a single branch. Because CNN can extract features from smart contract sample data and have both good expressiveness and generalization capabilities, they can improve the vulnerability detection capability of hybrid networks [34]. So, one of the two branches of the CBGRU model proposed in this study chooses CNN as the deep learning model. In studies on hybrid networks, CNNs are also mostly used in the feature extraction process of models due to their powerful feature extraction capabilities; for example, the CNN-SVM classifier proposed by Gong W et al. [35] and the DAQFF model proposed by Du S et al. [27]. The network model used in the feature extraction process of the other branch of the hybrid network proposed in this paper is the BiGRU model because smart contracts are context-dependent at runtime. GRU is a variant of RNN. Like LSTM, GRU is proposed to solve the gradient-vanishing problem that occurs in RNN, but GRU deals with the gradient-vanishing problem differently from LSTM. Compared with LSTM, GRU adds an update gate to replace the forgetting gate and input gate in LSTM, which greatly simplifies the results and parameters of the network and thus improves the training speed of the model. Furthermore, according to the study of Junyoung Chung et al. [36], GRU outperforms LSTM in some areas, and this paper also demonstrates experimentally that GRU outperforms LSTM in smart contract vulnerability detection tasks. BiGRU can further enhance the model’s feature-extraction capability in the feature-extraction phase. Therefore, the structure of the hybrid model proposed in this paper is shown in Figure 5.

As shown in Figure 5, the CBGRU hybrid model in this paper is composed of a total of three layers, namely, the word embedding layer, the feature extraction layer, and the classification layer. The word embedding model used in the first branch of the hybrid model is word2vec. Mikolov et al. proposed the word vector method word2vec can control the feature vector dimension and solve the dimensional catastrophe problem without ignoring the relative position relationship of phrases in the text and preserving the semantic relationship of phrases [37]. Word2Vec has two training modes, CBOW and Skip Gram, and the one used in this study is CBOW, which predicts intermediate words by context [38]. The use of Word2Vec in combination with CNN has been widely used in previous research, and word2vec is also one of the most popular word embedding methods that have been widely used in previous research, so word2vec is used in the first branch of the hybrid neural network proposed in this paper. The word embedding model used in the second branch is FastText, which is a variant form of word2Vec and differs from Word2vec in that the central word of the CBOW model is replaced with a category label. Finally, the model concatenates the features obtained from the two branches and performs vulnerability detection on the input smart contract. When the smart contract C is input to the model, the feature-learning process of the CBGRU hybrid model at moment *t* can be expressed as follows:(1)$$Word2Vec\left(C\right)\to C^{\omega }$$
(2)$$Fasttext\left(C\right)\to C^{\gamma }$$
(3)$$Convs\left(C^{\omega }\right)\to L_{t}$$
(4)$$BiGRU\left(C^{\gamma }\right)\to S_{t}$$
(5)$$Concat\left(L_{t},S_{t}\right)\to O_{t}$$
where $L_{t}$ denotes the features extracted by the CNN at moment *t* and $S_{t}$ denotes the features extracted by BiGRU at moment *t*. After that, feature fusion is performed to obtain the output of the CBGRU model at moment *t*. The CBGRU model detects the smart contract vulnerability process, as shown in Algorithm 1.
**Algorithm 1** Smart contract vulnerability detection process.**Input:***S*: Smart contracts that need to be tested**Output:***result*: the result of detection1: Step1. Use the preprocessing function *P* to preprocess the smart contract *S* to obtain $s_{p}$2: $S_{p}$ = P(S)3: Step2. Embedding $S_{p}$ using word2vec to obtain the embedding matrix $M_{1}$4: $M_{1}$ = $Word2vec\left(S_{p}\right)$5: Step3. Embedding $S_{p}$ using *FastText* to obtain the embedding matrix $M_{2}$6: $M_{2}$ = $FastText\left(S_{p}\right)$7: Step4. *CNN* performs feature extraction on $M_{1}$ to obtain features $F_{CNN}$8: $F_{CNN}$ = $CNN\left(M_{1}\right)$9: Step5. *BiGRU* performs feature extraction on $M_{2}$ to obtain features $F_{CBGRU}$10: $F_{BiGRU}$ = $BiGRU_{back}\left(M_{2}\right)$ + $BiGRU_{forward}\left(M_{2}\right)$11: Step6. Fusion of extracted feature values12:$\;F_{Hybrid}$ = $F_{CNN}⊕F_{BiGRU}$13: Step7. Classification by softmax to obtain results14: *result* = Softmax (W$F_{Hybrid}$ + *b*)

### 3.2. Word Embedding Layer

The main function of the word embedding layer is to process the original smart contract at the character level into the matrix, thus conforming to the input of the neural network. We collected all the keywords of solidity from Ethereum before the word embedding process, such as bool, break, case, catch, const, only owner, return, assert, event, indexed, union, etc. This is due to the smart contract vulnerability associated with the solidity keyword. For example, in the case of the reentrancy vulnerability, the attacker mainly uses the features of the call.value function of the Ethereum smart contract to achieve reentrant. We have processed the smart contract as follows.

Remove the solidity code version, such as “pragma solidity^0.4.4” in the ProofExistence contract in Figure 6.Removes comments, non-ASCII values, and blank lines from the contract.Represent user-defined function names as FUN plus numbers, and user-defined variable names as VAR plus numbers in smart contracts. This is because user-defined function names and variable names have little effect on whether the smart contract contains vulnerabilities and also add noise when performing feature extraction, which negatively affects the final feature extraction.Remove all spaces from the smart contract and perform word embedding; after the smart contract is processed, only the keywords in the solidity language will remain.

The process is shown in Figure 6.

As shown in Figure 6, the smart contract has been processed a total of three times. The first process is to remove the parts of the smart contract components that are not relevant to the vulnerability. The second process is to use each line of the previously processed smart contract as a fragment. The third process is to collect each character fragment in each line as a token, and the tokens are collected to generate a matrix form conforming to the deep learning neural network through the word embedding method. In this study, the processing is the same for both Word2Vec and FastText.

### 3.3. CBGRU Hybrid Network Layer

This subsection will introduce the CBGRU model proposed in this paper in detail. The layer structure of CBGRU is shown in Figure 7.

As can be seen in Figure 7, after the smart contract has been completed by the Word2Vec and Fastest models for word embedding, the two deep learning models start to perform feature extraction. In this branch of the CNN model, CNN uses the one-dimensional convolutional layer, because the size of the matrix generated after the smart contract has gone through the word embedding layer is (300, 100). After the data passes through the first convolution layer, it goes through a pooling layer to compress the number and parameters and reduce overfitting. The formula for the calculation of the one-dimensional convolutional layer in the model can be expressed as
(6)$$c_{j}^{n}=\underset{i}{\sum }X_{i}^{n-1}*w_{ij}^{n}+b_{j}^{n}$$
(7)$$X_{j}^{n}=ReLU\left(c_{j}^{n}\right)$$
(8)$$X_{n}^{j}=Flatten\left(X_{j}^{n}\right)$$
(9)$$X_{k}^{n+1}=FC\left(w_{kj}^{n+1}x_{j}^{n}+b_{k}^{n+1}\right)$$
where *w* represents the filter and *b* represents the offset; these symbols are the same as below. $X_{i}^{n-1}$ in Equation (6) represents the input of the convolution layer. We use the *ReLU* function as the activation function in the convolution calculation where $c_{j}^{n}$ denotes the output vector of the convolution layer. The *n* in the formula indicates the layers involved. In this study, a total of two convolutional layers are used for local feature learning. The next layer learns a nonlinear representation based on the output of the previous layer, and then feeds the learned representation to the next layer, forming a layered feature representation. *Flatten* in the formula represents the Flatten layer, and *FC* represents the fully connected layer, whose role is to reduce the dimension of the final output vector.

The deep learning model in another branch of CBGRU is BiGRU. Smart contracts first go through the Fastest model word embedding and then BiGRU for feature extraction. Since the size of the data is (300, 100), the *unit* of BiGRU is set to 300. After the feature extraction is completed, the activation operation is performed by the ReLU function, and the dropout layer is added to prevent overfitting. The forward propagation equation of the classical GRU model is expressed as
(10)$$r_{t}=\sigma \left(W_{r}\cdot \left[h_{t-1},X_{t}\right]\right)$$
(11)$$Z_{t}=\sigma \left(W_{z}\cdot \left[h_{t-1},X_{t}\right]\right)$$
(12)$$\tilde{h_{t}}=tanh\left(W_{\tilde{h}}\cdot \left[r_{t}*h_{t-1},x_{t}\right]\right)$$
(13)$$h_{t}=\left(1-Z_{t}\right)*h_{t-1}+Z_{t}*\tilde{h_{t}}$$
(14)$$y_{t}=\sigma \left(W_{o},h_{t}\right)$$

The $Z_{t}$ and $r_{t}$ in the formula represents the update gate and reset gate, respectively, and [] represents that the two vectors are connected. $\tilde{h_{t}}$ is represented as the candidate set at moment t, where *X* is the same as represented in the one-dimensional convolutional neural network as the input to the neural network. The *W* in the formula denotes the parameter to be learned. The BiGRU is added to the original one-way feedback as two-way feedback, and the BiGRU is capable of iterative processing of data in two directions. The equations for BiGRU can be expressed as
(15)$$\vec{r_{t}}=\sigma \left(\vec{W_{r}}\cdot \left[\vec{h_{t-1}},\vec{X_{t}}\right]\right)$$
(16)$$\vec{Z_{t}}=\sigma \left(\vec{W_{z}}\cdot \left[\vec{h_{t-1}},\vec{X_{t}}\right]\right)$$
(17)$$\vec{\tilde{h_{t}}}=tanh\left(\vec{W_{\tilde{h}}}\cdot \left[\vec{r_{t}}*\vec{h_{t-1}},\vec{x_{t}}\right]\right)$$
(18)$$\vec{h_{t}}=\left(1-\vec{Z_{t}}\right)*\vec{h_{t-1}\;}+\vec{Z_{t}}\;*\;\vec{\tilde{h_{t}}}$$
(19)$$\vec{y_{t}}=\sigma \left(\vec{W_{o}},\vec{h_{t}}\right)$$
(20)$$\overset{\leftarrow }{Z_{t}}=\sigma \left(\overset{\leftarrow }{W_{z}\cdot }\left[\overset{\leftarrow }{h_{t-1}},\overset{\leftarrow }{X_{t}}\right]\right)$$
(21)$$\overset{\leftarrow }{\tilde{h_{t}}}=tanh\left(\overset{\leftarrow }{W_{\tilde{h}}}\cdot \left[\overset{\leftarrow }{r_{t}}*\overset{\leftarrow }{h_{t-1}},\overset{\leftarrow }{x_{t}}\right]\right)$$
(22)$$\overset{\leftarrow }{h_{t}}=\left(1-\overset{\leftarrow }{Z_{t}}\right)*\overset{\leftarrow }{h_{t-1}\;}+\overset{\leftarrow }{Z_{t}}*\overset{\leftarrow }{\tilde{h_{t}}}$$
(23)$$\overset{\leftarrow }{y_{t}}=\sigma (\overset{\leftarrow }{W_{o}},\overset{\leftarrow }{h_{t}})$$
(24)$$y_{t}=\vec{y_{t}}*\overset{\leftarrow }{y_{t}}$$
where the arrows in two directions indicate the forward and backward processes, respectively, and the final node output at moment t in the hidden layer is $y_{t}$. After the feature extraction of the two branches of CBGRU’s smart contracts is completed, feature fusion is then performed through the connection layer. The feature fusion uses the concat method, and the fused feature matrix will be more adequate than the feature matrix extracted from a single network. Finally, the softmax layer will be used for classification to obtain the final result.

### 3.4. CBGRU Model Overall Process

The dataset used in this study can be expressed as $D=\left\{C_{1},\;C_{2},\;C_{3},\ldots .\;C_{m}\right\}$ after preprocessing is completed, where m represents a value of 42,569, which is the sum of samples in the dataset, $C_{i}$ is represented as a contract and $C_{i}$ is composed of k tokens that can be expressed as $C_{i}=\left\{x_{i1},x_{i2}\ldots .,x_{ik}\right\}$, where *x* is denoted as a token. In the first branch, the input data needs to be processed by word2vec to be transformed into input data suitable for CNN, and each token has to be embedded by word2vec to become $x_{il}^{\omega }$, denoting the *l*-th embedding vector of the *i*-th smart contract. After word2vec processing, the smart contract is transformed from $C_{i}$ to $C_{i}^{\omega }$. The composition of $C_{i}^{\omega }$ can be expressed as $C_{i}^{\omega }=\left\{x_{i1}^{\omega },\;x_{i}^{\omega },\;x_{\omega }^{\omega }\ldots ,x_{ik}^{\omega }\right\}$, then $C_{i}^{\omega }$ is input to CNN for feature extraction. The CNN model mainly includes a pooling layer and convolutional layer, which performs convolutional operations on data at different scales to eventually produce more complex features. The CNN model implementation process used in this paper is shown in the following equation.
(25)$$F_{CNN_{i}}=\int \left(w\cdot x_{i:i+s-1}^{\omega }+b\right)$$
(26)$$F_{CNN}=\left[F_{CNN_{1}}F_{CNN_{2}}F_{CNN_{3}}\ldots .,F_{CNN_{k-s+1}}\right]$$
where *w* denotes a filter for the CNN whose role is to generate a new feature value by convolution, s denotes the step size of the filter, *b* denotes the offset of the CNN, $F_{CNN_{i}}$ denotes a feature obtained by convolution operation, and $F_{CNN}$ denotes the feature vector obtained after pooling. In the second branch, the input data are processed by the FastText word embedding model, and the token in the original smart contract is transformed into $x_{il}^{\gamma }$, which represents the *l*-th embedding vector of the *i*-th smart contract. The deep learning model used in the second branch is BiGRU, which consists of a forward GRU (forward GRU) and a back GRU (back GRU). The BiGRU model implementation process used in this paper is shown in the following equation.
(27)$$F_{GRU_{tf}}=\int \left(F_{GRU_{tf-1}},x_{t}^{\gamma }\right)$$
(28)$$F_{GRU_{tb}}=\int \left(F_{GRU_{tb-1}},x_{it}^{\gamma }\right)$$
(29)$$F_{BiGRU_{t}}=F_{GRU_{tf}}+F_{GRU_{tb}}+b_{t}=\left(F_{GRU_{1}},\;F_{GRU_{2}},\;F_{GRU_{3}}\ldots \;F_{GRU_{t}}\right)$$
where $GRU_{tf}$ represents the hidden input of the forward GRU, $GRU_{tb}$ represents the hidden input of the back GRU, $x_{t}^{\gamma }$ denotes the output of GRU at moment t, and $b_{t}$ denotes the offset vector at time *t*. The features extracted by the two neural networks are fused to obtain the final features extracted by the CBGRU model. The features extracted by CBGRU are expressed in the following equation.
(30)$$F_{Hybrid}=F_{CNN}\;⊕\;F_{BiGRU}$$

The obtained hybrid features will be used as the input to the softmax layer of the model and the results will be obtained after classification.
(31)$$result=softmax\left(WF_{Hybrid}+b\right)$$
where *W* represents the weight matrix and *b* represents the offset matrix.

During the training of the CBGRU model, the main purpose is to improve the accuracy of the CBGRU model for smart contract vulnerability detection. The smart contract vulnerability detection task is also essentially a binary classification task, and the CBGRU model uses two different deep learning networks for feature extraction, and the two branches of the deep learning network are constantly learning during training, with the model trained to minimize the loss function. The CBGRU model uses the focal loss for calculating the loss, which is formulated as follows.
(32)$$Loss_{1}=-\frac{1}{n}\overset{n}{\underset{i=1}{\sum }}\alpha {\left(1-y_{i}^{\prime }\right)}^{\zeta }\log y_{i}^{\prime }+\left(1-\alpha \right)y_{i}^{\prime }{}^{\zeta }\log(1-y_{i}^{\prime })$$

The use of focal loss is mainly to solve the problem of sample imbalance and to reduce the weight of a large number of negative samples in training. In the formula, $y_{i}^{\prime }$ is the probability that the *i*-th sample is predicted to be positive, and ζ is the adjustment factor, whose the main function is to adjust the loss contribution of simple samples, paying special attention to hard-to-classify samples, and reduce the impact of simple samples. α is a balancing factor to balance the proportion of positive and negative samples and adjust their significance. In this study, along the lines of [39], default values (ζ = 2 and α = 0.25) were used. The training process of the CBGRU model is shown in Algorithm 2.
**Algorithm 2** training model.1: Initialize model parameters randomly2: Set the max number of epochs: $epoch_{max}$3: Set the origin dataset: *D*4: **for**
*S* in *D* **do**5:  // Use the preprocessing function *P* to process the processing smart contract *S*6:  *P(S)*7: **end for**8: for t in 1, 2, 3…, T **do**9:  Pack the dataset t into mini-batch: $D_{t}$10: **end for**11: **for** epoch in 1,2,3…,$epoch_{max}$12:  //Merge all datasets.13:  *D* = $D_{1}$
∪
$D_{2}\ldots \cup$
$D_{T}$14:  **for** $b_{s}$ in D **do**15:     $F_{CNN}$ = *CNN* ($Word2vec\left(b_{s}\right)$)16:     $F_{BiGRU}$ = *BiGRU* ($FastText\left(b_{s}\right)$)17:    *result* = *Softmax* ($F_{CNN}$ + $F_{BiGRU}$)18:    *Loss (*θ*)* = Equation (**32**)19:    Compute gradient: ▽(θ)20:    Update model: θ = θ − ε▽(θ)21:    **end for**22: **end for**

The hybrid model CBGRU proposed in this paper differs from the models in previous studies in that we combine different word embedding methods with different distinctive deep learning methods to obtain better results. The CBGRU model is the first to apply hybrid networks to smart contract vulnerability detection while achieving good detection results. In the following section, we present the experimental part of this paper to demonstrate the superiority of the performance of the hybrid model proposed in this paper through a large number of experiments.

## 4. Experiments and Results

In this subsection, we first introduce the performance metrics that we used in our experiments. The optimizer chosen for our study is Adam, which is used to update and compute the network parameters that affect the model training and model output to approximate or reach the optimal value. We chose Adam as the optimizer because Adam combines the performance of AdaGrad and RMSProp [40]. Adam provides optimization of methods for solving sparse matrix and noise problems and has been widely used in deep learning applications in recent years, especially for computer vision and natural language processing tasks. Referring to the currently popular TensorFlow [41] and Keras, the learning rate of the Adam optimizer was set to 0.001. The dropout of the Dropout layer in the CBGRU model was set to 0.5. This is because the randomly generated network structure is the most when the dropout is set to 0.5, which is beneficial for enhancing the generalization of the model. We set the epoch to 50 and the batch size to 128 in our experiments.

The experiments in this paper are divided into two parts: the first part is to discuss different deep learning models and word embedding models to prove the correctness of the CBGRU model proposed in this study, and the second part is to demonstrate the good performance of our proposed CBGRU model by comparing it with the models proposed in previous researches. In addition, we used different performance metrics, such as accuracy (ACC), precision (PRE), F1-score (F1), and false-positive rate (FPR), to evaluate the performance of our models.

### 4.1. Dataset

The dataset used in this paper is the recently released SmartBugs Dataset-Wild [42], a large-scale dataset of smart contract vulnerabilities based on the Solidity language. This dataset contains 47,587 real and unique sol files, which contain a total of about 203,716 smart contracts. We labeled the dataset based on the research of Durieux T et al. [43]. Because smart contracts can call each other, we treat a sol file as a smart contract when labeling data. According to the research of Durieux T et al., the sol files in SmartBugs Dataset-Wild were finally divided into two categories: smart contracts with vulnerabilities and smart contracts without vulnerabilities. There are six types of vulnerabilities contained in vulnerable smart contracts; the smart contract vulnerability categories are Stack Call Depth Attack vulnerability (Callstack Depth Attack), Integer Overflow vulnerability (Integer Overflow), Integer Underflow vulnerability (Integer Underflow), Reentry vulnerability (Reentry), Timestamp Dependency vulnerability (Timestamp Dependency), and Transaction Ordering Dependence vulnerability (Transaction Ordering Dependence). The number of smart contracts with vulnerabilities is 35,151, and the number of smart contracts without vulnerabilities is 12,247. The distribution of the numbers in the dataset is given in Figure 8.

From Figure 8, we can see that the sample data of smart contracts with and without vulnerabilities are not evenly distributed, and if such a dataset is used it will lead to overfitting. Therefore, we also used the smart contract dataset published by Peng Qian et al. [11]. To compare with previous studies, we change the transaction order dependency vulnerability (Transaction Ordering Dependence) to the Infinite Loop vulnerability (Infinite Loop) in the dataset. The distribution of the number of vulnerabilities in the final dataset is shown in Figure 9.

It can be seen from Figure 6 that the distribution of the number of various vulnerabilities in the dataset used in this study is reasonable compared to the original dataset. The number of vulnerabilities is shown in Table 1.

Finally, we selected the same number of smart contracts that do not contain vulnerabilities to form the final smart contract dataset.

### 4.2. Experiment

#### 4.2.1. Comparative Experiments

The core idea of the CBGRU model proposed in this paper is to combine the advantages of the word embedding model with those of the deep learning model to improve the performance of the model. Therefore, among the word embedding models, we chose the currently popular word2vec and FastText models, and among the deep learning models, we chose CNN, LSTM, GRU, BiGRU, and BiLSTM. We compare the performance of different deep learning models combined with different word embedding methods under the same algorithm and parameters. Among the models compared, M1 to M9 are hybrid models, which are structurally consistent with the CBGRU model, and M11 to M14 are single-branch deep learning models, where M10-A and M15-B are the branches in our proposed hybrid neural network, respectively. We conducted several experiments on 15 models to demonstrate that the CBGRU model is the optimal choice for smart contract vulnerability detection.

Since deep learning networks change their parameters after each training, we recorded the data from multiple tests of the model and chose the best one. In the experiment, each model was trained 50 times. The performance metric in the self-comparison experiment is the accuracy of the test set. The final results are shown in Table 2.

From Table 2, we can see that the CBGRU model proposed in this paper has better performance than the hybrid model proposed in this paper for data containing multiple smart contract vulnerabilities. The comparison between M11-A and M13 can be concluded that the combination of CNN and Word2Vec can obtain better detection performance in smart contract vulnerability detection. Similarly, the comparison between M11 and M12 shows that under the same conditions, BiGRU is more adequate for the extraction of smart contract vulnerability features. The comparison of M12 and M14 shows that the combination of BiGRU and FastText can achieve better feature extraction. Comparing M2 with M5, we can conclude that the bi-directional network is better than the normal network in feature extraction of smart contract vulnerabilities, and comparing M3 with M4, we can also get the same conclusion. Comparing M9 and CBGRU, it can be concluded that the CBGRU model proposed in this paper has good performance. From Table 2, we can see that M15-B has an excellent performance in single-structured networks. We also conducted experiments with M15-B as two branches of the hybrid model. We also experimented with M15-B as two branches of the hybrid model, but a gradient explosion occurred during the training period, indicating a problem with the structure of the network. The training process of the CBGRU model proposed in this paper is shown in Figure 10.

As can be seen from Figure 10, all four curves of the model gradually flatten out after the training count reaches 30 (epoch ≥ 30). Moreover, during the training process, the curves in the validation and training sets have the same trend and are close to each other, indicating that there is no overfitting in the model during the training. In the training process shown in Figure 10, the results of the model show an accuracy of 85.80%, recall of 86.18%, precision of 85.54%, and F1 of 85.86%. In this subsection, we experimentally demonstrate that the combination of CBHRU models proposed in this study is reasonable.

#### 4.2.2. Comparison with Previous Studies

In the previous subsection, we justified the CBGRU model by comparing it with different hybrid models, and this subsection will compare it with the models proposed in previous studies to demonstrate that the CBGRU model has good performance in smart contracts vulnerability detection. Since the methods proposed in the previous study target different vulnerabilities, the CBGRU model proposed in this paper is also tested for different smart contract vulnerabilities to compare the performance in smart contract vulnerability detection. The performance metrics used are accuracy, precision, recall, and F1. The CBGRU model proposed in this paper is trained and tested for six common smart contract vulnerabilities, namely, Infinite Loop, Timestamp Dependency, Integer Overflow, Reentry, Callstack Depth Attack, and Integer Underflow. The training process is shown in Figure 11.

From Figure 11, it can be concluded that the CBGRU model proposed in this paper has good detection performance for all six smart contract vulnerabilities. The test results for the six different vulnerabilities are shown in Table 3.

As can be seen from the table, for the four vulnerabilities of infinite loop, re-entry, timestamp dependency, and call stack depth attack, the CBGRU model proposed in this paper achieves accuracy and an F1-score of over 90%. However, both Integer Underflow and Integer Overflow have accuracy and an F1-score around 85%, which is lower than the remaining four vulnerabilities. The reason for this situation is that the features of these two vulnerabilities are not obvious in the code, so the correct rate is low. To demonstrate the superiority of the performance of the CBGRU model, we chose the models proposed in previous studies to compare the detection performance in the case of the same vulnerability. In the same experimental setting, we chose to compare the DeeSCVHunter model proposed by Yu X et al. [22], the *Peculiar* model proposed by Wu H et al. [23], the BLSTM-ATT model proposed by Peng Qian et al. [11], the TMP and DR-GCN models proposed by Zhang et al. [44], and the AME model proposed by Liu Z et al. [45]. Three different smart contract vulnerabilities were selected for comparison, and the results of the comparison are shown in Figure 12, Figure 13 and Figure 14.

As can be seen in Figure 12, the CBGRU model performs well in the smart contract reentry vulnerability detection task, with a precision of 96.30%, recall of 85.95%, an accuracy of 93.30%, and an F1-score of 90.02%. Peculiar has a precision of 91.80%, an accuracy of 92.37%, a recall of 92.40%, and an F1-score of 92.10%, which is the best performance among the models involved in the comparison. This was followed by DeeSCVHunter, with a precision of 90.70%, a recall of 83.46%, an accuracy of 93.02%, and an F1-score of 86.87%. Furthermore, the BLSTM-ATT has a precision of 88.50%, a recall of 88.48%, an accuracy of 88.47%, and an F1 score of 88.26%. The AME has a precision of 86.25%, a recall of 89.69%, an accuracy of 90.19%, and an F1 score of 87.94%. The DA-GCN has a precision of 89.84%, a recall of 82.00%, an accuracy of 91.15%, and an F1 score of 85.43%. The TMP has resulted in performance with a precision of 74.06%, a recall of 82.63%, an accuracy of 84.48%, and an F1-score of 83.82%. Finally, the obtained results portrayed that the DR-GCN has demonstrated poor performance by offering the lowest precision, 72.36%; the lowest recall, 80.89%; the lowest accuracy, 81.47%; and the lowest F1-score, 76.39%. The performance of TMP and DR-GCN is lower than the rest of the models and performs poorly. Since the F1-score is used to measure the values of precision and recall, we discuss the models involved in the comparison in terms of both accuracy and F1-score. From the comparison results, it can be seen that the accuracy of the CBGRU model proposed in this paper is higher than that of DeeSCVHunter, Peculiar, BLSTM-ATT, AME, and DA-GCN. When performing a single smart contract vulnerability detection test in this paper, the number of smart contract samples containing vulnerabilities and the number of samples without smart contract vulnerabilities in the dataset is the same, so the higher the correct rate indicates the better the performance of the deep learning network when performing vulnerability detection. Among the models involved in the comparison, the BLSTM-ATT uses the word2vec model for word embedding and feature extraction via the BLSTM model. The BLSTM-ATT model uses a single word embedding approach and a single deep learning model. The CBGRU model proposed in the paper combines the advantages of different word embedding models and deep learning models, and it can be seen from the comparison results that the accuracy of the CBGRU model is 4.49% higher than that of BLSTM-ATT and the F1-score is 2.66% higher than that of the BLSTM-ATT. Among all the models involved in the comparison, the models with an F1-score higher than 90% are the CBGRU model and the Peculiar model, and when the F1-score is high, it indicates that the vulnerability detection method used in the model is efficient. The F1-score of Peculiar is slightly higher than the F1-score of the CBGRU model proposed in this paper. In performance metrics, Peculiar has a higher recall rate but lower precision rate than the CBGRU model. It means that Peculiar can detect more samples containing smart contract vulnerabilities than the CBGRU model, while the CBGRU model can identify more samples containing smart contract vulnerabilities than Peculiar in the detection results. The reason is that Peculiar is a detection model designed for smart contract reentry vulnerabilities. Peculiar also uses critical data flow graph (CDFG) techniques for pre-processing; so, Peculiar only uses critical information related to reentry vulnerabilities to detect whether a smart contract contains a reentry vulnerability. The Peculiar model reduces the interference of useless information in vulnerability detection information by using CDFG. The CBGRU model proposed in this paper is capable of detecting multiple vulnerabilities. The CBGRU model reduces the impact of irrelevant information during smart contract vulnerability detection by extracting smart contract keywords, replacing custom variables, and removing parts that are irrelevant to vulnerability detection when pre-processing smart contract samples. However, different smart contract vulnerabilities are associated with different keywords, and when the CBGRU model performs the reentry vulnerability detection task, the processed smart contract samples still contain the keywords used to detect other vulnerabilities, thus affecting the extraction of vulnerability features by the CBGRU model. This results in a slightly lower F1-score for the CBGRU model than for the Peculiar model.

As can be seen in Figure 13, the CBGRU model has an accuracy of 93.02%, a recall of 97.45%, a precision of 89.47%, and an F1-score of 93.29% in the timestamp vulnerability detection task. From the comparison results, it can be seen that the CBGRU model outperforms the rest of the models in the timestamp vulnerability detection task. Among the models involved in the comparison, the DA-GCN model uses a single GCN model for smart contract vulnerability feature extraction, while the CBGRU model uses a BiGRU model and CNN model for feature extraction. The CBGRU model is 5.48% higher than DA-GCN in terms of accuracy and 8.46% higher than DA-GCN in terms of F1-score, which can be concluded that the feature values extracted by combining the advantages of different deep learning networks are more adequate. From Figure 14, it can be seen that the CBGRU model proposed in this paper performs very well in the infinite loop vulnerability detection, with an accuracy of 93.16%, recall of 89.15%, precision of 98.29%, and F1-score of 93.50%; it also can be seen that CBGRU model is higher than the rest of the models in terms of performance metrics. Compared to the AME model, which performed well in the comparison model, the CBGRU model improved 12.84% in accuracy and 14.62% in F1-score.

From the above discussion, it can be seen that the CBGRU model proposed in this paper is higher than the rest of the models in terms of accuracy in the smart contract reentry vulnerability detection task, and only the F1-score was slightly lower than the F1-score of the Peculiar model. The CBGRU model performs well in both the timestamp vulnerability detection task and the infinite loop vulnerability detection task. From Figure 14, we can see that the CBGRU model outperforms other models in detecting timestamp vulnerabilities detection and in finite loop vulnerabilities detection. In the test results of all three vulnerabilities of smart contracts, the CBGRU model proposed in this paper maintains an above 90% F1 score. In summary, the CBGRU model proposed in this study has good performance for smart contract vulnerability detection.

On the one hand, the CBGRU model has higher accuracy in vulnerability detection tasks and is capable of detecting multiple vulnerabilities compared to previous studies. On the other hand, the smart contract vulnerability detection model proposed in this paper also has limitations. First, our proposed CBGRU model can only detect whether a smart contract contains a vulnerability, and cannot identify the type of vulnerability for a smart contract that contains multiple vulnerabilities. Second, our proposed CBGRU model is capable of detecting multiple smart contract vulnerabilities, but the accuracy of detecting smart contract vulnerabilities with distinct features is higher than that of smart contract vulnerabilities with insignificant features. For example, in the Integer Overflow vulnerability detection task, CBGRU has an accuracy of 86.54%, but in the Infinite Loop vulnerability detection task, CBGRU has an accuracy of 93.16%. Finally, deep learning models are more effective on huge datasets, but the dataset used in this study is not very large.

## 5. Conclusions

In this paper, we propose a deep learning-based hybrid network model CBGRU. The presented model operates on four stages, namely, smart contract pre-processing, word embedding, feature extraction, and classification. We completed the word embedding of the smart contracts using the FastText and Wrod2Vec models. In one branch of the CBGRU model, the features are extracted from the Word2vec embedding using CNN, whereas, in another branch, features are obtained from FastText embedding using BiGRU. In the classification phase, the features obtained from the two branches are merged and transferred to the softmax layer for classification.

We conducted two different main experiments in order to verify the performance of the CBGRU model proposed in this paper. We use a publicly available dataset in our experiments. In the first experiment, we created 15 basic deep learning models by combining different deep learning models with different word embedding models. We experimentally demonstrate that our proposed CBGRU model outperforms other basic models in performing smart contract vulnerability detection. In the second experiment, we chose three different smart contract vulnerabilities for comparison purposes. The results of the comparison show that CBGRU model has accomplished maximum performance with an accuracy of 93.30%, 93.02%, and 93.16% in reentry vulnerability, timestamp vulnerability, and infinite loop vulnerability, respectively. The results show that the CBGRU proposed in this paper has a higher accuracy and better classification performance and that the CBGRU model is capable of detecting multiple smart contract vulnerabilities.

We experimentally demonstrate that the CBGRU approach has excellent smart contract vulnerability detection and higher accuracy than other models in testing tasks against a wide range of smart contract vulnerabilities. In addition, CBGRU can perform vulnerability detection in the local network, which is more convenient and faster than traditional smart contract vulnerability detection tools. In this paper, we also experimentally demonstrate that using different word embedding methods enables the model to extract feature values more adequately. In the follow-up work, we will try to identify multiple smart contract vulnerabilities in the same smart contract and improve the accuracy of the CBGRU model to detect vulnerabilities in smart contracts with obscure features.

## Figures and Tables

**Figure 1 sensors-22-03577-f001:**
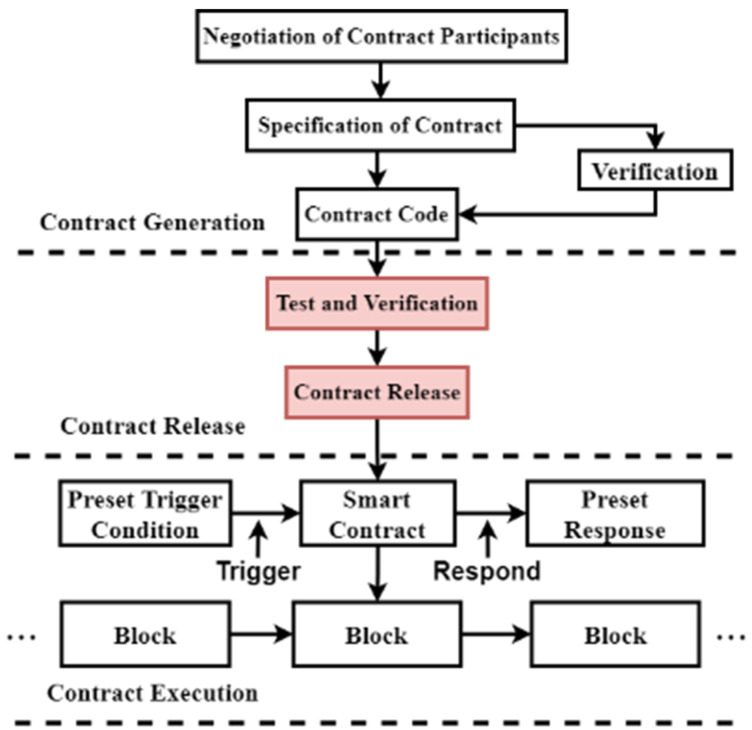
The life cycle of a smart contract.

**Figure 2 sensors-22-03577-f002:**
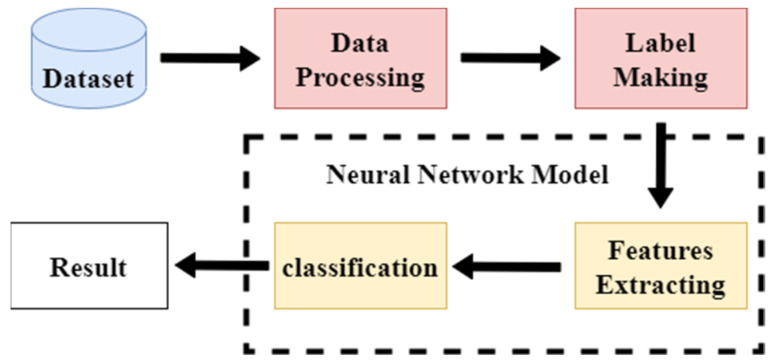
The single model classification process.

**Figure 3 sensors-22-03577-f003:**
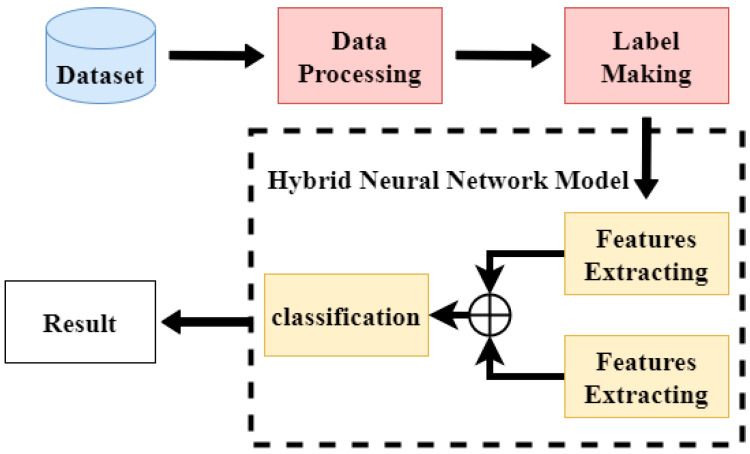
The hybrid model classification process.

**Figure 4 sensors-22-03577-f004:**
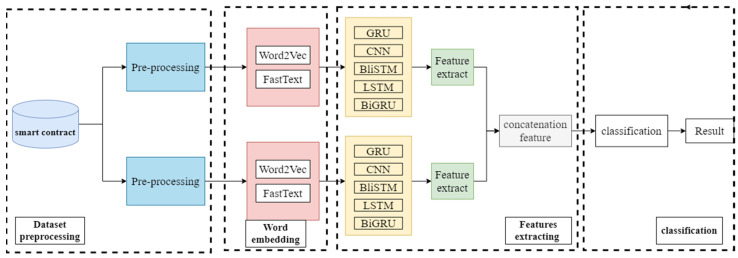
Hybrid network overall framework in this paper.

**Figure 5 sensors-22-03577-f005:**
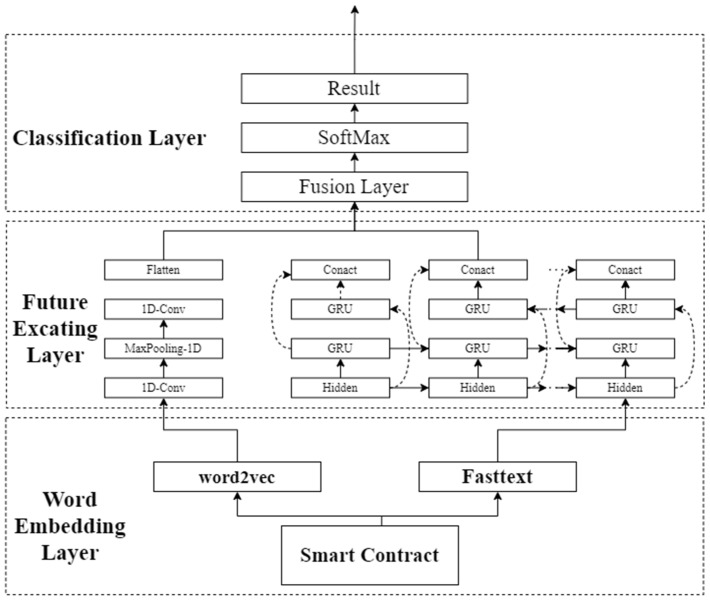
CBGRU hybrid model structure.

**Figure 6 sensors-22-03577-f006:**
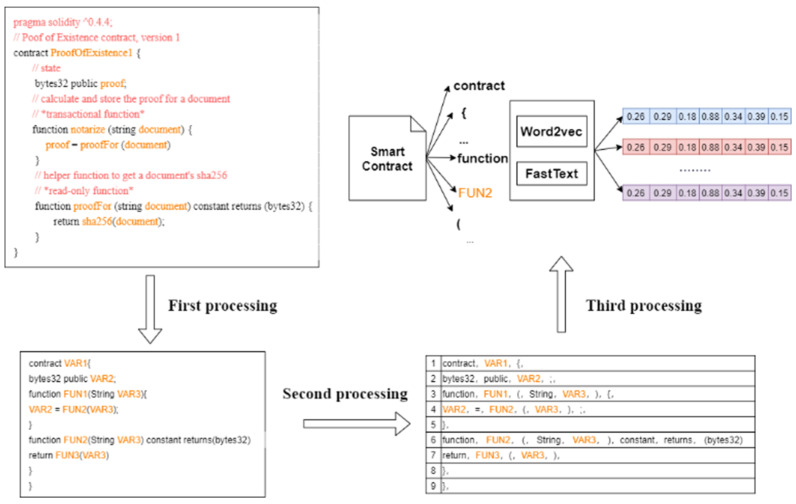
Smart contract word embedding process.

**Figure 7 sensors-22-03577-f007:**
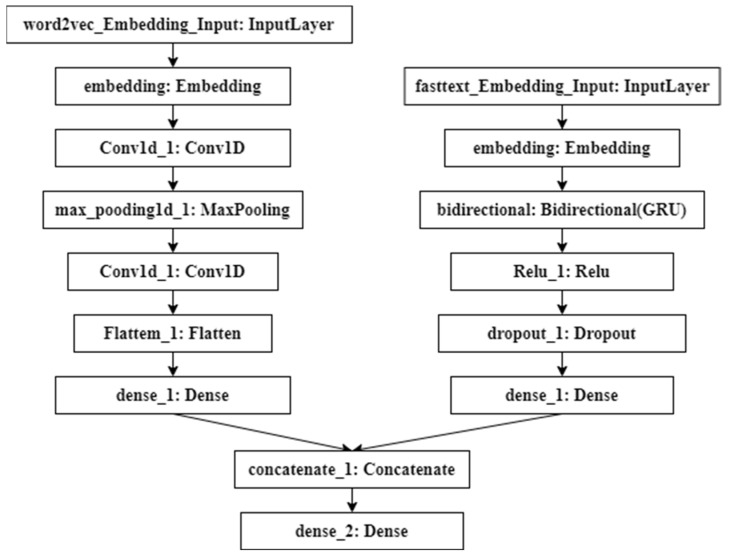
CBGRU layer structure.

**Figure 8 sensors-22-03577-f008:**
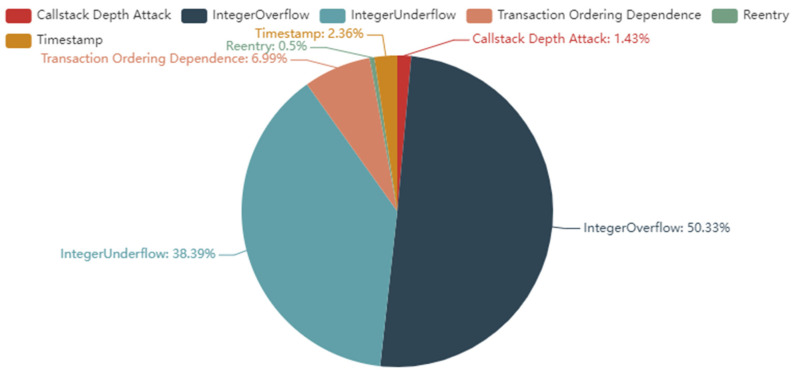
Dataset distribution.

**Figure 9 sensors-22-03577-f009:**
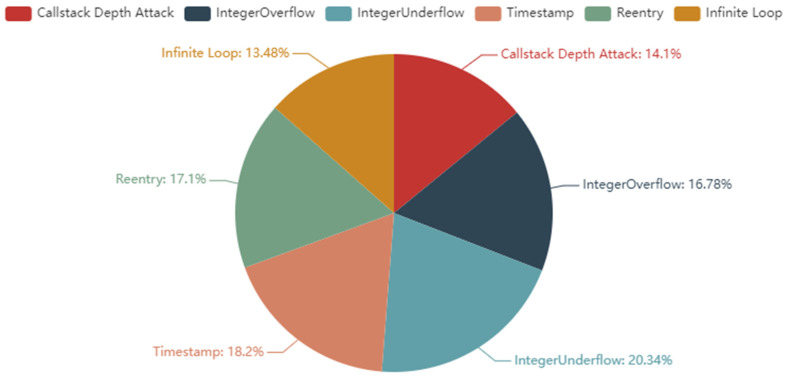
Distribution of the number of vulnerabilities.

**Figure 10 sensors-22-03577-f010:**
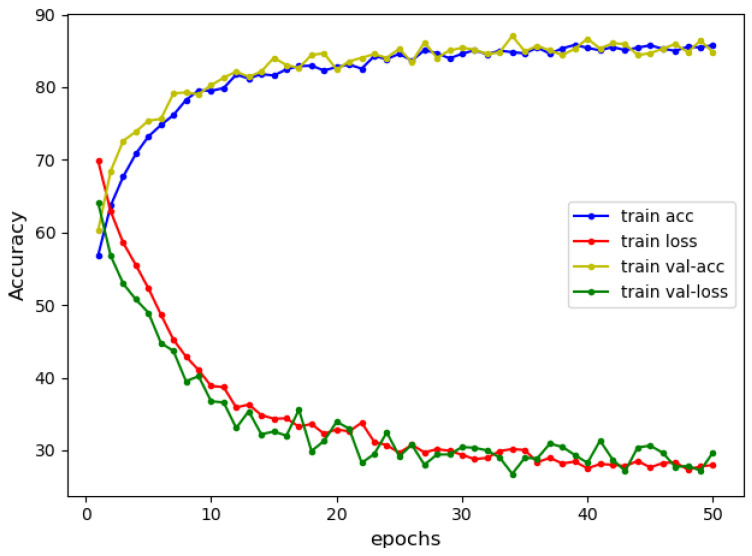
CBGRU training process.

**Figure 11 sensors-22-03577-f011:**
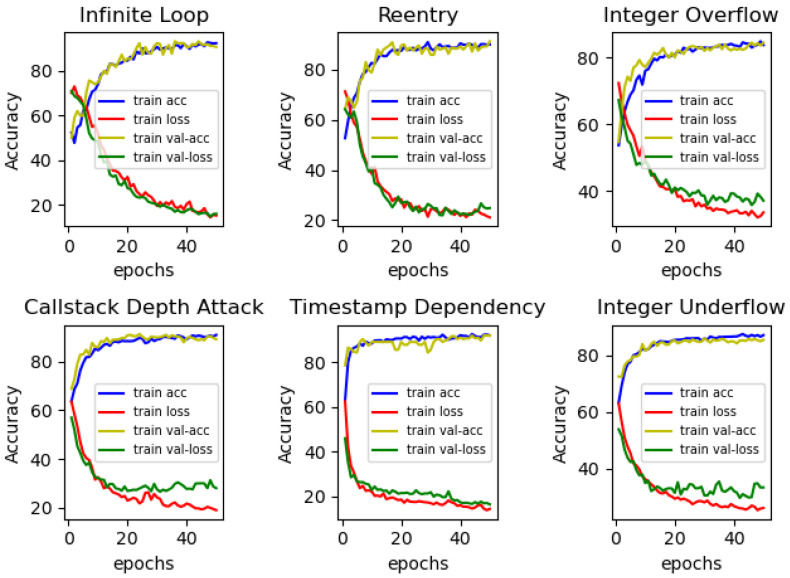
CBGRU model training process.

**Figure 12 sensors-22-03577-f012:**
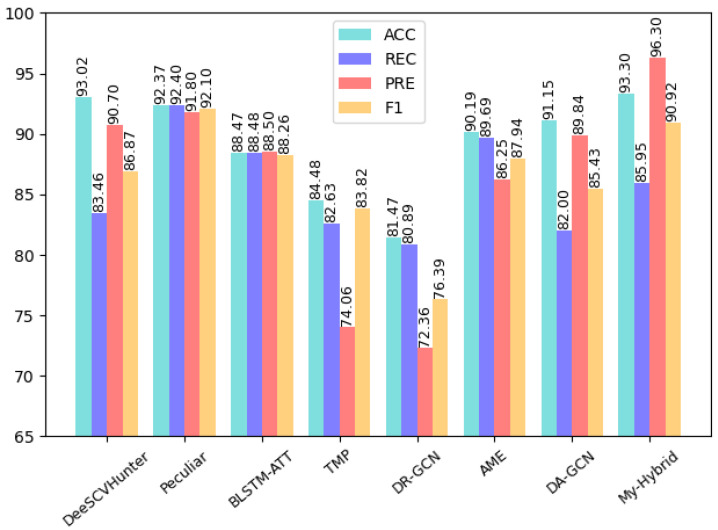
Reentry vulnerability detection results.

**Figure 13 sensors-22-03577-f013:**
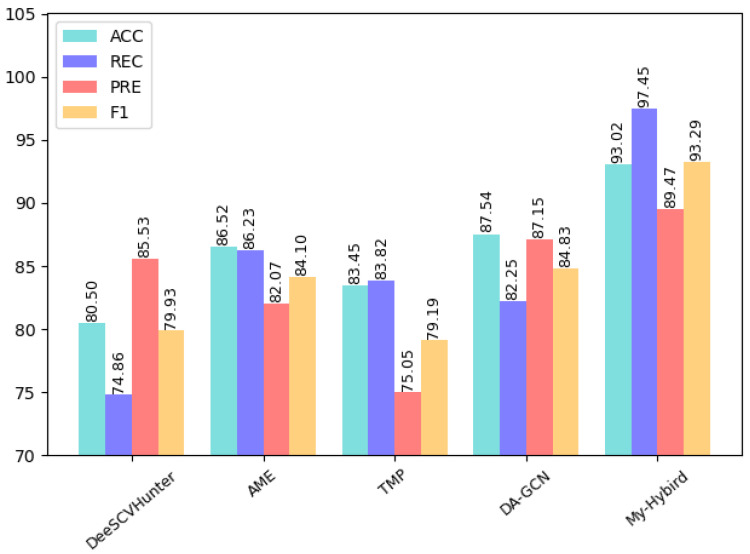
Timestamp vulnerability detection results.

**Figure 14 sensors-22-03577-f014:**
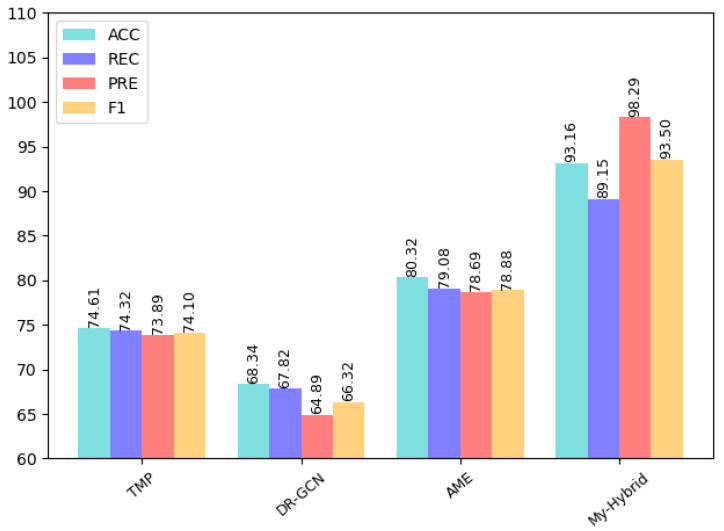
Infinite loop vulnerability detection results.

**Table 1 sensors-22-03577-t001:** Number of smart contract vulnerabilities.

Vulnerability Name	Numbers
Callstack Depth Attack	1378
Integer Overflow	1640
Integer Underflow	1988
Reentry	1719
Timestamp Dependency	1671
Infinite Loop	1317

**Table 2 sensors-22-03577-t002:** Self-comparison model.

Model Name	Classification Method One	Embedding Method	Classification Method Two	Embedding Method	Embedding Size	Dropout	Epoch	Accuracy (%)
M1	CNN	Word2Vec	CNN	Word2vec	300	0.5	50	80.78
M2	CNN	Word2Vec	GRU	Word2vec	300	0.5	50	80.33
M3	CNN	Word2Vec	LSTM	Word2vec	300	0.5	50	79.67
M4	CNN	Word2Vec	BiLSTM	Word2vec	300	0.5	50	81.14
M5	CNN	Word2Vec	BiGRU	Word2vec	300	0.5	50	82.10
M6	CNN	Word2Vec	CNN	FastText	300	0.5	50	79.25
M7	CNN	Word2Vec	GRU	FastText	300	0.5	50	80.67
M8	CNN	Word2Vec	LSTM	FastText	300	0.5	50	79.45
M9	CNN	Word2Vec	BiLSTM	FastText	300	0.5	50	83.55
CBGRU	CNN	Word2Vec	BiGRU	FastText	300	0.5	50	85.80
M10-A	CNN	Word2Vec	\	\	300	0.5	50	75.67
M11	BiLSTM	Word2Vec	\	\	300	0.5	50	74.60
M12	BiGRU	Word2Vec	\	\	300	0.5	50	75.56
M13	CNN	FastText	\	\	300	0.5	50	77.65
M14	BiLSTM	FastText	\	\	300	0.5	50	75.04
M15-B	BiGRU	FastText	\	\	300	0.5	50	78.75

**Table 3 sensors-22-03577-t003:** Test results for the six different vulnerabilities.

Vulnerability Type	Accuracy	Precision	Recall	F1-Score
Infinite Loop	93.16%	89.15%	98.29%	93.50%
Reentrancy	93.30%	96.30%	85.95%	90.92%
Integer Overflow	86.54%	87.23%	85.66%	86.43%
Callstack Depth Attack	90.31%	90.04%	88.41%	90.21%
Timestamp Dependency	93.02%	89.47%	97.45%	93.29%
Integer Underflow	85.43%	86.15%	84.42%	85.28%

## Data Availability

The experimental data and associated code used in this study have been deposited in the GitHub repository (https://github.com/xiaoaochen/CBGRU) accessed on 31 March 2022.

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
