# Peer review of "CBGRU: A Detection Method of Smart Contract Vulnerability Based on a Hybrid Model"

_sensors, 2022, doi:10.3390/s22093577_

Round 1

Reviewer 1 Report

The manuscript tackles one of the interesting subjects in security, more specifically, the security of blockchain. It proposes a model to detect smart contract vulnerabilities. However, the authors should address the following comments when preparing the revised version:

  1. The authors claim in the abstract that “But in the real world, collecting contractual vulnerability data requires huge human and time costs. To solve these problems”. Does your model which built on a data set is going to solve these problems? I think you should re-consider such statements.
  1. There is no description of the benchmark database and I believe such a factor is critical in evaluation.
  2. The comparison results were presented in plots and simply re-stated in text, but there is no in-depth discussion.
  3. The authors should clearly justify the need for their approach in the presence of many effective approaches and should clearly state the pros and cons of their suggested technique.
  4. The author should use the following recent related references to update the introduction and the related work sections:
  • A. Alzubi, J. A. Alzubi, K. Shankar, D. Gupta, “Blockchain and artificial intelligence enabled privacy‐preserving medical data transmission in Internet of Things”, Transactions on Emerging Telecommunications Technologies, Vol. 32, No. 12, 2022, pp. e4360-e4374.
  1. The conclusion should be abstracted and self-contained, so authors need to consider re-drafting it.

Reviewer 2 Report

This papar proposes a deep learning moduel to detect vulnerability in smart contract. The contents are explaned clear and the structure is well organized. However, I suggest some minor comments before the publication.

  1. I suggest the authors to make a comparsion of the proposed work and the existing studies in Related Work section.
  2. I suggest the authors to add some pesudo codes of the proposed algorithm in section 3 to give a better understanding.
  3. Some recent works can be cited in the Introduction section to improve the theoretical depth of this paper.
    1. Lei Hang, BumHwi Kim, DoHyeun Kim, "A Transaction Traffic Control Approach Based on Fuzzy Logic to Improve Hyperledger Fabric Performance", Wireless Communications and Mobile Computing, vol. 2022, Article ID 2032165, 19 pages, 2022. https://doi.org/10.1155/2022/2032165

Round 2

Reviewer 1 Report

I recommend that the authors check the manuscript again for linguistic and grammatical errors.